# Outcomes of a bedside ultrasound-guided peripherally-inserted central catheter placement across critically-ill older patients

Kyungwon Lee[1,2☯], Kyoung Won Yoon📧[3☯], Minchang Kang[4], Donghyoun Lee📧[5]*

1 Department of Surgery, Korea University Guro Hospital, Seoul, South Korea, 2 Department of Surgery, Armed Forces Capital Hospital, Seongnam, South Korea, 3 Department of Surgery, Chung-Ang University Gwangmyeong Hospital, Gwangmyeong, South Korea, 4 Department of Critical Care Medicine, H Plus Yangji Hospital, Seoul, South Korea, 5 Department of Surgery, Jeju National University Hospital, Jeju National University School of Medicine, Jeju, South Korea

☯ These two authors contributed equally to the work.
* dlee@jejunu.ac.kr

## Abstract

### Background

An ultrasound (US)-guided peripherally inserted central catheter (PICC) is a thin, flexible tube inserted into a vein in the upper arm and then guided into a large vein near the heart, using US for precise vein location. We conducted this single-center, retrospective study to describe outcomes of a bedside US-guided PICC across critically-ill older patients in a single small-volume center in an intensive care unit (ICU) setting in Korea.

### Methods

We included 452 Korean older ICU patients aged ≥60 years who received PICC at our hospital between January of 2021 and December of 2024. A logistic regression analysis with odds ratio (OR) was performed to identify risk factors of the non-optimal position of catheter tip. The overall PICC-related infection-free survival was expressed as mean±standard error, for which 95% confidence intervals (CIs) were provided and the statistical significance was analyzed using the log-rank test. The corresponding Kaplan-Meier survival and hazards were plotted as a curve.

### Results

There were a total of 13 cases (2.88%) of the PICC-related infection. Of these, there were five cases (1.1%) of PICC-related bloodstream infection. A total of 421 patients (93.1%) had optimal positions of the PICC tip. A logistic regression analysis showed that male sex (OR 0.202; 95% CI 0.078–0.521, $p = 0.001$), the length of a catheter (OR 0.786; 95% CI 0.683–0.904, $p = 0.001$) and right side (OR 4.415; 95% CI 1.649–11.824, $p = 0.003$) were significant risk factors of non-optimal positions of the catheter.

**Data availability statement:** All relevant data are within the manuscript and its Supporting Data files.

**Funding:** This study was supported by a grant from the Jeju National University Hospital Research Fund (Grant No. 2019-35). But the funder had no role in study design, data collection and analysis, decision to publish or preparation of the manuscript.

**Competing interests:** The authors have declared that no competing interests exist.

Time-to-events are estimated at 56.29 ± 0.98 days (95% CI 54.37–58.21). Moreover, survival rates are estimated to reach 0.918 ± 0.032 (95% CI 0.858–0.983) at 31 days of the PICC use.

## Conclusions

We describe outcomes of a bedside US-guided PICC across critically-ill older patients in a single small-volume center in an ICU setting in Korea.

---

## Introduction

The peripherally-inserted central catheter (PICC) is defined as a medical device for central vascular access and it is inserted in the extremity and then advanced until its tip is positioned in the vena cava [1]. It is a medium-to-long-term venous access that is placed through a peripheral vein, such as basilic vein, brachial vein or cephalic vein, serving as a standard, cost-effective alternative to the central venous catheter (CVC) [2,3].

Despite the benefits of PICC, complications associated with its use have been described in the literature, thus termed as PICC-related complications. These include thrombophlebitis, catheter-related infection, particularly termed as the PICC-related infection, and thrombosis and mechanical complications (*e.g.*, occlusion and accidental withdrawal), all of which cause patient discomfort and additional health care costs [4–10]. Of these, the PICC-related infection, defined as a significant concern associated with the use of PICC. Such cases of infection may arise when bacteria or other germs enter the catheter and bloodstream, potentially leading to serious complications, is a less common complication as compared with serious thrombosis [11]. According to a review of literatures, the incidence of PICC-related infection is estimated at 0.4–0.8 per 1,000 catheter-days, although a majority of these cases are seen in a lower risk group of patients [12–17]. Moreover, severe cases of PICC-related complications include life-threatening bloodstream infection (BSI) and deep vein thrombosis (DVT) [18,19]. It has been shown that the presence of a PICC line is closely associated with a high risk of PICC-related BSI [6,20].

According to a previous study, bedside ultrasound (US)-guided PICC was a feasible, safe modality in intensive care unit (ICU) patients even in a single small-volume center in the pandemic era of the COVID-19 [21]. Along the continuum of this previous literature, we have devised an evidence-based protocol to control the bedside PICC-related infection in older adults ICU patients receiving bedside US-guided PICC in a small-volume center. The older adults are vulnerable to an increase in the frequency and duration of hospitalization and often have multiple comorbidities. This justifies the importance of an effective, reliable venous access in the treatment of the older adults with geriatric diseases [22]. We therefore conducted this single-center, retrospective study to describe outcomes of a bedside US-guided PICC across critically-ill older patients in a single small-volume center in an ICU setting in Korea.

## Materials and methods

### Study patients and setting

A total of 512 patients received PICC at a 21-bed ICU of our 291-bed medical institution between January of 2021 and December of 2024.

We included Korean older ICU patients aged ≥60 years.

Exclusion criteria for the current study are as follows:

(1) The patients aged between 20 and 29 years old (n = 5)

(2) The patients aged between 30 and 39 years old (n = 7)

(3) The patients aged between 40 and 49 years old (n = 14)

(4) The patients aged between 50 and 59 years old (n = 31)

(5) The patients lost to follow-up (n = 3).

We therefore included a total of 452 patients in the current study, whose age distribution is shown in Fig 1; it was conducted in compliance with the relevant ethics guidelines following the approval by the Institutional Review Board (IRB) of our medical institution (IRB approval #: HYJ 2022-07-017). All procedures described herein were performed in accordance with the 1964 Declaration of Helsinki and its later amendments or comparable ethical standards. But informed consent was waived due to its retrospective nature.

We reported the study findings in accordance with the STROBE checklist [23].

### An evidence-based protocol to perform bedside US-guided PICC and to control its related infection

We used US guidance for PICC placement can reduce the number of attempts and mechanical complications, but ensured that it should only be performed by those with full training [24]. In our series, bedside US (GE LOGIQ-e Ultrasound Machine; GE Healthcare, Milwaukee, WI)-guided PICC was performed, as previously described (Fig 2) [21]. All the procedures were performed using a 5-Fr Dual Lumen Power PICC (Bard Access System Inc., Salt Lake City, UT, USA). Lack of timely control of catheter-related infection would lead to an increase in the incidence of further infection and

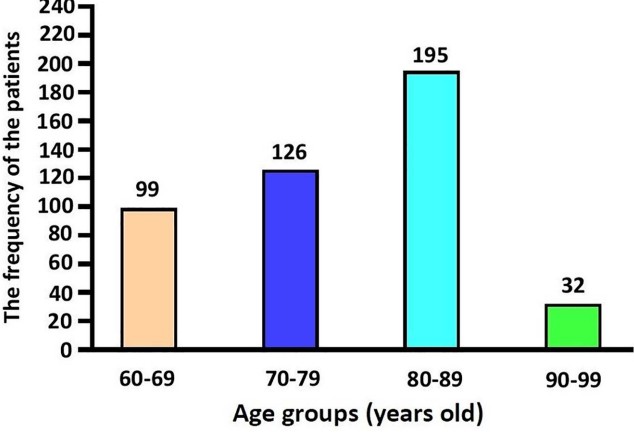

**Fig 1. Age distribution of the eligible patients.**

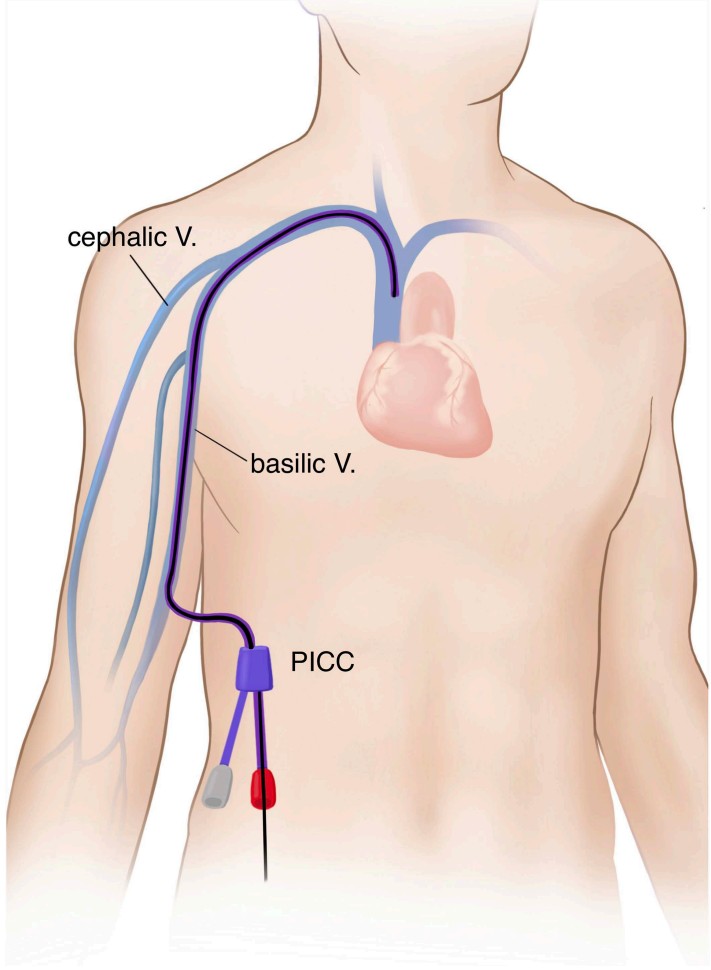

**Fig 2. A peripherally-inserted central catheter (PICC). Note:** V., vein.

mortality [25]. It would therefore be mandatory to identify methods for reducing or eliminating the bedside PICC-related infection.

The patients were placed in a supine position. But the patients with dyspnea were placed in a sitting position with the arm abducted, and they had a tourniquet applied to the upper arm. The patients underwent PICC placement under maximal barrier precautions, including the use of a surgical cap, sterile gown, sterile gloves, and large sterile drapes. For skin antisepsis, we used 2% chlorhexidine in 70% isopropyl alcohol. Thus, the procedure was performed using the microintroducer technique at the patients' bedside under US guidance [26–28].

To determine the side of PICC insertion, the site without previous catheterization was selected. Moreover, the site on the right arm was initially selected for PICC insertion in the patients with no past history of catheterization. For puncture, large-sized veins were selected so that they might not be less compressed during the procedure under US guidance. Therefore, brachial and basilic veins rather than cephalic veins were selected [29–31].

The length of catheter was measured based on a sum of the distance from the insertion site to the axilla, that from the axilla to the sternum and that from the sternum to the 4th intercostal space [32]. After the completion of PICC, the internal jugular vein (IJV) was scanned under US guidance to ensure the correct placement of the catheter. This was followed by chest X-ray to confirm the location of the catheter [33].

After the procedure, the patients were evaluated on routine surveillance at a weekly follow-up and the PICC was daily flushed using a 10 mL saline [34].

## Patient evaluation and criteria

Baseline characteristics of the patients include age, sex, Simplified Acute Physiology Score II (SAPS2), status at the time of ICU arrival (ventilation and continuous renal replacement therapy [CRRT]), length of ICU stay, length of hospital stay, reasons for ICU discharge (transfer to a ward or other hospitals and death) and reasons for hospital discharge (discharge to home, transfer to other hospitals and death).

Causes of ICU admission include central nervous system problems, cardiovascular system problems, pulmonary diseases, hepatobiliary problems, gastrointestinal problems and renal problems.

PICC-related characteristics of the patients include purposes of PICC placement, number of previous catheters, side of PICC placement, length of PICC, veins for PICC placement, length of procedure time, length of indwelling time, rates of PICC-related infections and reasons for PICC removal.

The position of the catheter tip was classified into the optimal position, the suboptimal position and the malposition. The optimal position was defined as the placement of catheter tip in the areas extending from the distal 2/3 of the superior vena cava (SVC) to the right atrium (RA). Moreover, the suboptimal position was defined as the placement of catheter tip into tributaries of the SVC connecting to the RA. Finally, the malposition was defined as the placement of catheter tip into other veins than the abovementioned ones. For the PICC insertion, the suboptimal position was maintained but the malposition was corrected [35,36].

The time-to-event (TTE) is defined as the length of period between a certain time of origin and the time of the occurrence of the event of interest [37]. That is, the TTE is referred to as time to the length of time between the onset of the PICC-related infection and the PICC removal. Moreover, the survival of PICC is defined as its maintenance without removal because of the PICC-related infection, as previously described [34,37–41].

## Statistical analysis

All data was expressed as mean±standard deviation or the number of the patients with percentage, where appropriate. Continuous and categorical variables were analyzed using the Student's $t$-test and $\chi^2$-test, respectively. Moreover, a logistic regression analysis with odds ratio (OR) was performed to identify risk factors of the non-optimal position of catheter tip. Before multiple regression analysis, multicollinearity was resolved by confirming variance inflation factors. The overall PICC-related infection-free survival was expressed as mean±standard error, for which 95% confidence intervals (CIs) were provided and the statistical significance was analyzed using the log-rank test. The corresponding Kaplan-Meier survival and hazards were plotted as a curve. A $p$-value of $<0.05$ was considered statistically significant. All statistical analyses were performed using the IBM SPSS Statistics Version 28.0.1.1 (IBM, Armonk, NY).

## Results

### Baseline and clinical characteristics of the patients

From the study population, 60 patients aged 59 years or younger were excluded from the current study (Fig. 3). Therefore, a total of 452 patients were finally included in it, who comprise 244 men (54.0%) and 208 women (46.0%) and whose mean age was 77.89±8.67 (60–99) years old. The age distribution of the eligible patients showed that there were 195 patients (43.1%) in their 80s, 126 (27.9%) in their 70s, 99 (21.9%) in their 60s and 32 (7.1%) in their 90s. Baseline characteristics of the patients are represented in Table 1.

A total of 299 patients (66.2%) received the PICC for long-term intravenous access, 318 (70.4%) had no past history of receiving catheters and 263 (58.2%) received the PICC on the right side. Target veins include basilic vein (203 (44.9%)), brachial vein (178 (39.4%)) and cephalic vein (71 (15.7%)) (Table 1).

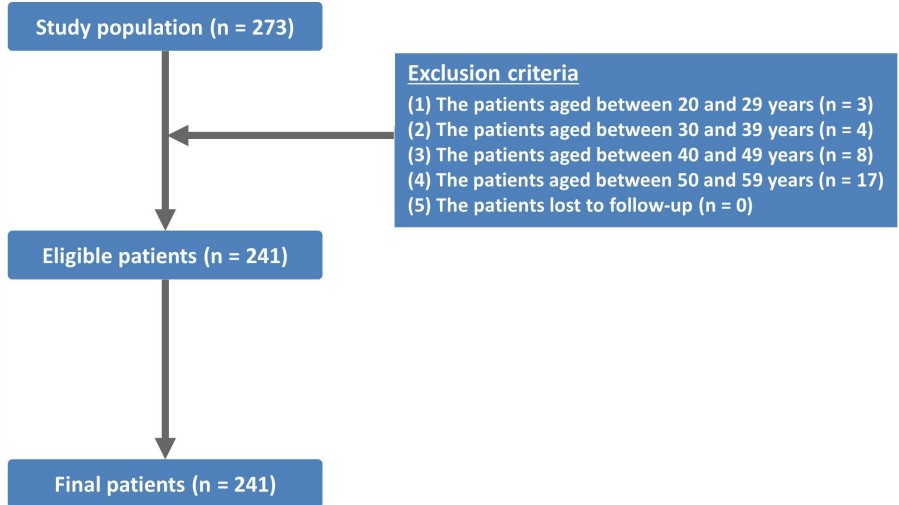

**Fig 3. Disposition of the study patients.**

## Cases of the PICC-related infection

In our series, there were a total of 13 cases (2.88%) of the PICC-related infection. Of these, there were five cases (1.1%) of PICC-related BSI (Table 2).

## Locations of the PICC tip and risk factors of its non-optimal position

In our series, a total of 421 patients (93.1%) had optimal positions of the PICC tip (Table 1). A logistic regression analysis showed that male sex (OR 0.202; 95% CI 0.078–0.521, $p=0.001$), the length of a catheter (OR 0.786; 95% CI 0.683–0.904, $p=0.001$) and right side (OR 4.415; 95% CI 1.649–11.824, $p=0.003$) were significant risk factors of non-optimal positions of the catheter (Table 3).

## Cumulative PICC-related infection-free survival and hazards

In our series, TTEs are estimated at $56.29\pm0.98$ days (95% CI 54.37–58.21) (Table 4). Moreover, survival rates are estimated to reach $0.918\pm0.032$ (95% CI 0.858–0.983) at 31 days of the PICC use (Table 5). The corresponding Kaplan-Meier curves are plotted in Fig. 4 and 5.

## Discussion

Since the early 1990s, a PICC has been used in a clinical setting. It remains problematic, however, that adverse effects may occur at a higher incidence during intra-hospital transport in critically-ill patients [42,43]. This justifies the rationale of performing the bedside PICC placement for them [44,45]. In 2001, Lee DS, et al. described the results of a preliminary report about the US-guided PICC performed by an intensivist in an ICU setting in a large-volume center in Korea [46].

In 1989, the FDA published a precautionary statement as to the positioning of CVCs to the effect that the catheter tip cannot be placed in or migrate into the heart [47]. In 1998, the National Association of Vascular Access Networks published a statement about the positioning of PICCs, thus recommending that the tip of a PICC should be located within the lower 1/3 of the SVC, close to the SVC-RA junction [48].

It is a general rule that the PICC is blindly inserted based on anatomical measurements of estimated distance [49]. The conventional use of anatomical landmarks provides the estimated length between the puncture site and the SVC-RA

**Table 1. Baseline characteristics of the patients (n = 452).**

| Variables | Values |
|---|---|
| **Age (years old)** | 77.89 ± 8.67 (60-99) |
| 60-69 | 99 (21.9%) |
| 70-79 | 126 (27.9%) |
| 80-89 | 195 (43.1%) |
| 90-99 | 32 (7.1%) |
| **Sex** | |
| Men | 244 (54.0%) |
| Women | 208 (46.0%) |
| **SAPS2** | 39.27 ± 14.45 |
| Ventilator | 182 (40.3%) |
| CRRT | 79 (17.5%) |
| **Status** | |
| **Length of ICU stay (days)** | 24.38 ± 27.79 |
| **Length of hospital stay (days)** | 41.04 ± 42.71 |
| **Causes of ICU admission** | |
| **CNS disorders** | |
| Brain hemorrhage | 30 (6.6%) |
| Brain infarction | 18 (4.0%) |
| Infectious diseases | 5 (1.1%) |
| **CVS disorders** | |
| HF | 52 (11.5%) |
| ACS | 20 (4.4%) |
| PTE | 4 (0.9%) |
| Infectious diseases | 4 (0.9%) |
| AD | 4 (0.9%) |
| **Pulmonary system disorders** | |
| Pneumonia | 185 (40.9%) |
| COPD | 7 (1.5%) |
| Tuberculosis | 4 (0.9%) |
| Lung cancer | 11 (2.4%) |
| **Hepatobiliary system disorders** | |
| Liver cirrhosis | 3 (0.7%) |
| rHCC | 6 (1.3%) |
| Spleen injury | 2 (0.4%) |
| Cholecystitis/Cholangitis | 22 (4.9%) |
| Pancreatitis | 2 (0.4%) |
| **Gastrointestinal system disorders** | |
| Bleeding | 12 (2.7%) |
| Perforation | 22 (4.9%) |
| Obstruction | 6 (1.3%) |
| IBD | 4 (0.9%) |
| **Renal system disorders** | |
| AKI | 29 (6.4%) |
| **Causes of ICU discharge** | |
| Transfer to a ward | 258 (57.1%) |
| Transfer to other hospital | 66 (14.6%) |

*(Continued)*

**Table 1.** (Continued)

| Variables | Values |
| --- | --- |
| Death of a patient | 128 (28.3%) |
| **Causes of hospital discharge** | |
| Return to home | 104 (23.0%) |
| Transfer to other hospital | 208 (46.0%) |
| Death of a patient | 140 (31.0%) |
| **Purposes of PICC** | |
| Long-term intravenous access | 299 (66.2%) |
| Use of non-peripherally compatible infusate | 153 (33.8%) |
| **The number of previous catheters** | |
| 0 | 318 (70.4%) |
| ≥ 1 | 134 (29.6%) |
| **Side of PICC and the length of a catheter** | |
| Right side | 263 (58.2%) |
| Length (cm) | 38.53 ± 2.82 |
| Left side | 189 (41.8%) |
| Length (cm) | 43.52 ± 3.21 |
| **Target veins** | |
| Brachial vein | 178 (39.4%) |
| Basilic vein | 203 (44.9%) |
| Cephalic vein | 71 (15.7%) |
| **Length of procedure time (min)** | 14.20 ± 7.33 |
| **Length of catheter indwelling time (days)** | 16.56 ± 10.58 |
| **Causes of PICC removal** | |
| Improvement of a patient's condition | 172 (38.1%) |
| Death of a patient | 97 (21.5%) |
| Transfer to other hospital with PICC | 62 (13.7%) |
| Accidental removal | 16 (3.5%) |
| Replacement with new catheter | 56 (12.4%) |
| Suspicious catheter-related infection | 12 (2.7%) |
| Non-function | 9 (2.0%) |
| Swelling or hematoma | 21 (4.6%) |
| Malposition | 7 (1.5%) |
| **Locations of a catheter tip** | |
| **Optimal position** | 421 (93.1%) |
| **Suboptimal position** | |
| RA | 10 (2.2%) |
| Tributaries of the SVC | 10 (2.2%) |
| **Malposition** | |
| Ipsilateral IJV | 3 (0.7%) |
| Ipsilateral SCV | 2 (0.4%) |
| Arm vein | 4 (0.9%) |

Abbreviations: SAPS2, Simplified Acute Physiology Score II; CRRT, continuous renal replacement therapy; ICU, intensive care unit; CNS, central nervous system; CVS, cardiovascular system; HF, heart failure; ACS, acute coronary syndrome; PTE, pulmonary thromboembolism; AD, aortic dissection; COPD, chronic obstructive pulmonary disease; rHCC, ruptured hepatocellular carcinoma; IBD, inflammatory bowel disease; AKI, acute kidney disease; PICC, peripherally-inserted central catheter; RA, right atrium; SVC, superior vena cava; IJV, internal jugular vein; SCV, subclavian artery.

Values are mean±standard deviation with range or the number of the patients with percentage, where appropriate.

**Table 2. Summary of cases of the peripherally-inserted central catheter-related infections.**

| Variables | Values | | | |
|---|---|---|---|---|
| N | 13 (2.88%) | | | |
| Case # | Age/Sex | Length of catheter indwelling time (days) | Pathogens isolated from the tip culture | Pathogens isolated from the blood culture |
| 1 | 91/M | 28 | *Staphylococcus capitis* | Negative |
| 2 | 66/M | 28 | *Candida parapsilosis* | *Candida parapsilosis* |
| 3 | 84/F | 14 | *Streptococcus mitis, S. oralis* and *Corynebacterium species* | Negative |
| 4 | 66/M | 28 | *Staphylococcus epidermidis* | Negative |
| 5 | 78/F | 16 | *Stenotrophomonas maltophilia* | Negative |
| 6 | 72/M | 21 | *Enterococcus faecali* | *Enterococcus faecali* |
| 7 | 88/F | 18 | *Klebsiella pneumoniae* | Negative |
| 8 | 65/M | 29 | *Pseudomonas aeruginosa* | *Pseudomonas aeruginosa* |
| 9 | 79/M | 25 | MRSA | MRSA |
| 10 | 87/F | 19 | ESBL-producing *Escherichia coli* | Negative |
| 11 | 93/M | 17 | *Candida albicans* | *Candida albicans* |
| 12 | 74/F | 20 | *Acinetobacter baumannii* | Negative |
| 13 | 69/M | 22 | *Serratia marcescens* | Negative |

**Note:** N, the total number of cases; F, female; M, male.

**Abbreviations:** MRSA, methicillin-resistant *Staphylococcus aureus*; ESBL, extended-spectrum β-lactamase.

**Table 3. Risk factors of non-optimal positions of a catheter tip.**

| Variables | OR 95% CI | *p*-value |
|---|---|---|
| Age | 1.024 (0.976-1.074) | 0.330 |
| Male sex | 0.202 (0.078-0.521) | 0.001* |
| SAPS2 | 1.007 (0.978-1.037) | 0.628 |
| Long-term intravenous access | 0.665 (0.281-1.573) | 0.353 |
| Length of procedure time | 0.977 (0.917-1.04) | 0.462 |
| Length of a catheter | 0.786 (0.683-0.904) | 0.001* |
| Number of previous catheters | 1.994 (0.889-4.476) | 0.094 |
| Right side | 4.415 (1.649-11.824) | 0.003* |

**Note:** OR, odds ratio; CI, confidence interval.

**Abbreviation:** SAPS2, Simplified Acute Physiologic Score II.

*Statistical significance at $p < 0.05$.

**Table 4. Cumulative complication-free survival period.**

| The total number of the patients | Incidences of the PICC-related infection | Censored value | TTEs (days) |
|---|---|---|---|
| 452 | 9 | 433 (98.0%) | 56.29±0.98 (54.37-58.21) |

**Abbreviation:** PICC, peripherally-inserted central catheter; TTE, time-to-event.

TTEs are expressed as meanstandard error with 95% confidence intervals.

junction. Its advantages include relatively lower financial costs and acceptable performance. It can therefore be considered as the first-line of choice in patients with poor economic status. It is disadvantageous, however, in that patients are exposed to radiation and burdened with additional costs because of post-procedural chest X-rays [50]. Moreover, the

**Table 5. Cumulative hazards.**

| Time points | n1 | n2 | Survival rates |
|---|---|---|---|
| 13 | 270 | 1 | 0.996±0.004 (0.989-1.000) |
| 14 | 259 | 1 | 0.992±0.005 (0.982-1.000) |
| 16 | 226 | 1 | 0.988±0.007 (0.975-1.000) |
| 20 | 172 | 1 | 0.982±0.009 (0.965-1.000) |
| 22 | 145 | 1 | 0.976±0.011 (0.954-0.998) |
| 27 | 108 | 1 | 0.967±0.014 (0.939-0.995) |
| 28 | 95 | 1 | 0.956±0.017 (0.923-0.991) |
| 29 | 65 | 1 | 0.942±0.022 (0.899-0.987) |
| 31 | 40 | 1 | 0.918±0.032 (0.858-0.983) |

**Note:** n1, the total number of the patients; n2, incidence of the peripherally-inserted central catheter-related infection.

Survival rates are expressed as meanstandard error with 95% confidence intervals.

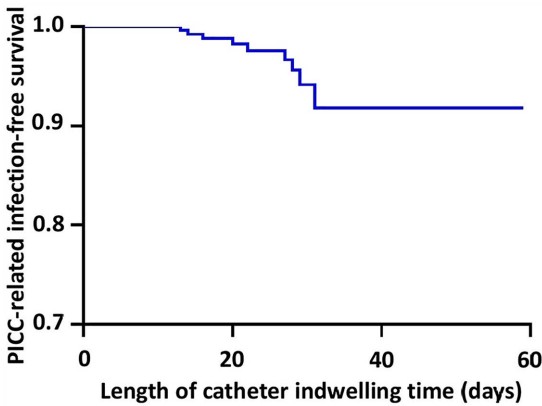

**Fig 4. Kaplan-Meier surivival. Abbreviation:** PICC, peripherally-inserted central catheter. In our series, time-to-events are estimated at 56.29±0.98 days (95% CI 54.37–58.21).

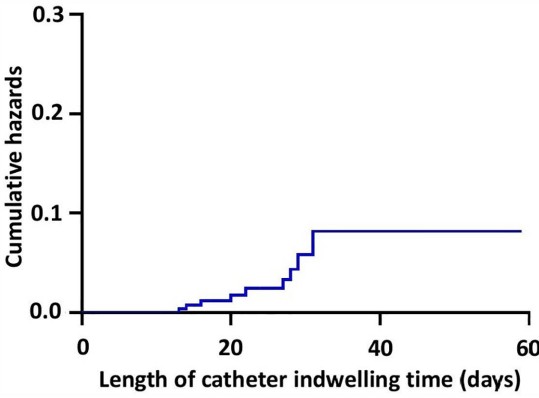

**Fig 5. Kaplan-Meier cumulative hazards.** Survival rates are estimated to reach 0.918±0.032 (95% CI 0.858-0.983) at 31 days of the peripherally-inserted central catheter use.

success rate of the PICC based on the conventional use of anatomical landmarks is estimated at approximately 80%, which may not be regarded as a satisfactory outcome [2,51,52]. This explains why we routinely perform the PICC under US guidance in our institution. The utility of US guidance in bedside PICC has been advocated by several previous reports [53–57]. The success rate of conventional PICC varies, ranging from 65% to 75%, because there is a limited availability of the site of PICC (1.5 inches above to 1.5 inches below the antecubital fossa). But the success rate of US-guided PICC increases to 91–94% [1]. This is in agreement with our results showing that the rate of optimal position was 93.1% (421/452).

Despite the efficiency of the PICC, it may be unavoidable that positioning of a PICC tip may cause complications; they can be classified as mechanical (obstruction) and organic (infections and/or thromboses) complications [58]. It would therefore be mandatory *not only* to position a PICC in the remote region where there is a sufficient blood flow with the endothelium far from the catheter *but also* to ensure that the tip cannot reach the RA, which is essential for minimizing a risk of such complications. In this regard, the SVC-RA junction serves as the optimal site of PICC [49,59]. A tip malpositioning may raise a risk of PICC-related complications, such as venous thrombosis, vascular erosion, malfunction or arrhythmias [51,60,61]. If recognized at the earliest opportunities possible, however, a tip malpositioning would be corrected and this would be helpful for preventing the occurrence of serious complications [60,61]. Special attention should therefore be paid to positioning of the catheter from the left side because a > 40°-angle formed between the PICC tip and the wall of the SVC may cause the perforation of the vascular wall and the pericardium covering the last portion of the SVC, infection, malfunctioning and venous thrombosis [59]. Additionally, vascular erosion and early cardiac tamponade may also occur as a result of accidental vessel/heart perforation during the PICC placement [62]. An accurate technical PICC procedure would therefore be useful in improving its overall performance [63].

Previous studies have indicated that the major disadvantage of PICC is its association with possible risks of DVT and pulmonary embolism [64–69]. This is not surprising because PICCs occupy a large portion of the cross-sectional diameter of peripheral veins of the upper extremities and are involved in the onset of venous stasis [70]. Moreover, they are vulnerable to displacement. Therefore, the displacement of the PICC tip can also cause damages to the endothelium [71]. Thus, the PICC-related DVT account for up to 35% of all cases of DVT in the upper extremities [72]. Risk factors of the PICC-related DVT include severity of illness, malignancy, a past history of taking warfarin, that of venous thrombosis or thromboembolism; high body mass index (BMI), trauma, renal failure, the infusion of antibiotics, the external diameter of the PICC >4 F, the left-sided placement of PICC and the placement of PICC in the basilic vein [64,72–75]. It is known that the degree of risk of DVT in the upper extremities is higher in association with the PICC as compared with the CVC [76]. This is supported by Winters JP, et al. who showed that patients receiving PICC are at a significantly greater risk of DVT in the upper extremities as compared with those receiving CVC (8.1 *versus* 4.8 per 1,000 admissions; $p < 0.05$) [77]. In this regard, it is promising that there were no cases of DVT in our series.

In our series, there were nine cases (2.0%) of PICC-related infections. This is in agreement with a previous published report that the incidence of PICC-related exit-site infections is estimated at 1.9–60.9% [11]. PICC-related infections have a strong relationship with a length of hospital stay, ICU status, the number of device lumens, a previous PICC placement, an operator's technical expertise, the time of catheter retention, the time of PICC indwelling, white blood cell counts, a history of diabetes mellitus and immunity [78–81]. Moreover, risk factors of PICC-related infections include patient-related factors (*e.g.*, chronic illnesses, immunosuppression and malnutrition) *as well as* catheter-related factors (*e.g.*, the time of PICC indwelling, catheter care delay and the site of PICC placement) [11].

In our retrospective cohort, the right-side PICC insertion for the patients with no prior PICC insertion involves inserting the catheter into a vein in the arm, typically the basilic, brachial or cephalic vein under US guidance, with the catheter advanced towards the SVC near the heart [32]. The right IJV is generally favored for PICC insertion due to its larger size and straighter path to the SVC, making it easier to access and potentially reducing complications. However, the left-sided IJV catheterization may be preferred in specific circumstances, and an operator's handedness may affect the technique

[82–84]. This is supported by peer-reviewed evidence based on anatomical advantages and lower rates of complications. In more detail, the right IJV is typically larger and has a more direct route to the SVC, making it easier to cannulate and potentially reducing a risk of complications, such as vessel injury or catheter malposition [84]. The right-sided IJV catheterization is associated with a lower rate of complications, such as carotid artery puncture, hematoma and pneumothorax as compared with the left-sided one [85–88].

Here, we describe our single-center, retrospective experience with bedside US-guided PICCs in older ICU patients in a small-volume center in Korea.

Limitations of the current study are as follows: First, we conducted it under retrospective design in a small series of the patients who had been admitted to an ICU of a secondary referral center. Therefore, the possibility of selection bias could not be completely ruled out. Fourth, there was no control group in the current study. Second, we failed to analyze the learning curve of bedside US-guided PICCs in ICU patients in a small-volume center. It remains difficult to improve skills in performing bedside US-guided PICCs in ICU patients [53]. Nevertheless, there is a paucity of literatures showing the learning curve for acquisition of this capacity, although a recent study showed that the PICC is a well-established procedure with a small learning curve [89,90]. This is supported by Lassen K, et al. showing that the safety and efficacy of PICC are achieved without a learning curve [91]. Moreover, according to Kleidon TM, et al., it would be mandatory consider the variability in the learning curve in employing a new medical device into a healthcare service [92]. Finally, the possibility of a potential bias arising from the right-sided preference in the patients with no prior PICC insertion could not be completely ruled out. It is likely that the patients with a history of PICC insertion may have had more severe underlying conditions or require more frequent infusions as compared with those without it. This causes differences in baseline health conditions and thereby could affect outcomes of the current study, with no respect to the PICC insertion line placement, which eventually makes it difficult to rule out differences in effects between the right-sided and left-sided cases [93].

## Conclusions

In conclusion, we describe outcomes of a bedside US-guided PICC across critically-ill older patients in a single small-volume center in an ICU setting in Korea. But further large-scale case-controlled studies are warranted to corroborate our results.

## Supporting information

**S1 File. Supporting information for the current study is the excel file of the patient raw data.**
(XLSX)

## Acknowledgments

We would like to thank Siwon Yoon from Ewha Womans University for visualization of the figures.

## Author contributions

**Conceptualization:** Kyungwon Lee, Minchang Kang, Donghyoun Lee.

**Data curation:** Kyoung Won Yoon, Minchang Kang.

**Formal analysis:** Kyoung Won Yoon, Minchang Kang, Donghyoun Lee.

**Funding acquisition:** Minchang Kang, Donghyoun Lee.

**Investigation:** Kyoung Won Yoon, Minchang Kang.

**Methodology:** Kyungwon Lee, Minchang Kang.

**Project administration:** Minchang Kang, Donghyoun Lee.

**Resources:** Kyungwon Lee, Minchang Kang.

**Software:** Minchang Kang.

**Supervision:** Kyungwon Lee, Minchang Kang, Donghyoun Lee.

**Validation:** Kyoung Won Yoon, Minchang Kang, Donghyoun Lee.

**Visualization:** Kyungwon Lee, Minchang Kang, Donghyoun Lee.

**Writing – original draft:** Kyoung Won Yoon, Minchang Kang, Donghyoun Lee.

**Writing – review & editing:** Kyoung Won Yoon, Minchang Kang, Donghyoun Lee.

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
