## [Editor Report · Decision Letter 0]

28 Mar 2025

Dear Dr. Lee,

**1. Abstract: Please provide a structured abstract, with headings such as Background, Methods, Results, and Conclusions, or as otherwise requested by the Journal. It is very difficult to understand the methods and background for this study from the abstract as it mainly focuses on results.**
**2. Introduction: The introduction is far too long (3.5 pages). This should be around 1 page, double spaced, with normal margins. Please make it more concise with a focus on relevant things without getting too detailed (i.e. the history of PICC, etc.). Other aspects can perhaps be placed in the Discussion if the authors feel is necessary. 3. Methods: Please place the Methods section directly after the Introduction and directly before the Results. 4. Methods: Please add the STROBE checklist and mention this in the Methods: https://www.strobe-statement.org/checklists/ 5. Methods: It is unclear why mean and SD were used. I am concerned that the data distribution may be skewed. Please either provide data to suggest otherwise or consider median and IQR, along with non-parametric statistical tests 6. Methods: As outlined in Table 5, please mention the exact regression technique in the methods (i.e. "logistic regression modelling was conducted...") along with these covariates and how they were selected. Importantly, were they checked for colinearity? Please also provide the univariable, unadjusted estimates in a separate column for each variable (i.e. row). 7. Tables: There are far too many tables. Please consider combining or place some in a Supplementary File, alongside the STROBE checklist (as outlined in point 4 above) 8. Figures: Please add risk tables with events/total Ns to Figures 1 and 2. As per STROBE checklist, please also add a patient flow diagram that shows how patients were selected/how many excluded/how many finally included. Figure 3 should also ideally be proportion rather than number, please consider having the y-axis as %. 9. The study would also benefit from a close read and extensive typographical edits. For instance: 1) "A total of 241 patients (n=241)". The (n=241) is unnecessary. 2) Rather than discussing granular age distributions, you can mention the mean or median (SD or IQR). 3) Figures 1 and 2 have no numerical data and so it is not sufficient to just refer to them in the text. Please provide some numbers in the actual results. 4) I am not sure why the results are all of a sudden summarized mid way through the Discussion. Ideally, the first paragraph of Discussion should provide a concise summary, then findings can be contextualized and discussed. 5) Limitations should be in one cohesive paragraph and each point should be expanded upon (i.e. retrospective design, leading to the potential for misclassification and/or residual confounding). There are several other instances, including capitalization/grammatical changes needed. 10. The title is far too long. Consider something like: *"Outcomes of a bedside central catheter placement protocol across critically ill older adults: A retrospective study"***

Please submit your revised manuscript as soon as possible or by May 12 2025 11:59PM. If you will need more time than this to complete your revisions, please reply to this message or contact the journal office at plosone@plos.org . A rebuttal letter that responds to each point raised by the academic editor. You should upload this letter as a separate file labeled 'Response to Reviewers'.A marked-up copy of your manuscript that highlights changes made to the original version. You should upload this as a separate file labeled 'Revised Manuscript with Track Changes'.An unmarked version of your revised paper without tracked changes. You should upload this as a separate file labeled 'Manuscript'.

We look forward to receiving your revised manuscript.

Kind regards,

Kiyan Heybati

Academic Editor

PLOS ONE

“This study was supported by a grant from the Jeju National University Hospital Research Fund (Grant No. 2019-35).”

3. For studies involving third-party data, we encourage authors to share any data specific to their analyses that they can legally distribute. PLOS recognizes, however, that authors may be using third-party data they do not have the rights to share. When third-party data cannot be publicly shared, authors must provide all information necessary for interested researchers to apply to gain access to the data. (https://journals.plos.org/plosone/s/data-availability#loc-acceptable-data-access-restrictions)

4) All necessary contact information others would need to apply to gain access to the data.

---

## [Decision Letter · Decision Letter 1]

15 Jun 2025

Dear Dr. Lee,

Thank you for submitting your manuscript to PLOS ONE. After careful consideration, we feel that it has merit but does not fully meet PLOS ONE’s publication criteria as it currently stands. Therefore, we invite you to submit a revised version of the manuscript that addresses the points raised during the review process.

We look forward to receiving your revised manuscript.

Kind regards,

Kiyan Heybati, MD MSc

Academic Editor

PLOS ONE

Reviewers' comments:

Reviewer's Responses to Questions

**Comments to the Author**

Reviewer #1: All comments have been addressed

Reviewer #2: (No Response)

Reviewer #3: All comments have been addressed

Reviewer #4: (No Response)

Reviewer #5: (No Response)

2. Is the manuscript technically sound, and do the data support the conclusions?

Reviewer #1: Yes

Reviewer #2: Partly

Reviewer #3: Partly

Reviewer #4: Yes

Reviewer #5: (No Response)

3. Has the statistical analysis been performed appropriately and rigorously?

Reviewer #1: Yes

Reviewer #2: I Don't Know

Reviewer #3: Yes

Reviewer #4: Yes

Reviewer #5: (No Response)

4. Have the authors made all data underlying the findings in their manuscript fully available?

Reviewer #1: Yes

Reviewer #2: Yes

Reviewer #3: Yes

Reviewer #4: Yes

Reviewer #5: (No Response)

5. Is the manuscript presented in an intelligible fashion and written in standard English?

Reviewer #1: Yes

Reviewer #2: Yes

Reviewer #3: Yes

Reviewer #4: Yes

Reviewer #5: (No Response)

Reviewer #1: 1-We therefore conducted this single-center, retrospective study to describe our protocol in a single small-volume center in an ICU setting in Korea. What do you mean protocol. What is the results of protocol. Protocol or outcome.

2- The patients underwent PICC placement under maximal barrier precautions using 2% chlorhexidine for skin preparation, surgical drapes, a surgical cap, sterile gown and gloves. 2% chlorhexidine alone or in combination with alcohol 70%.

3-In our series, there were a total of five cases (2.1%) of the PICC-related infection. What is the definition of PICC infection in your study.

Reviewer #2: The inclusion and exclusion criteria were not well described. Equally, the study question was not well described.

The objetives of the study should be defined so it can guide the statistical analysis and the data comparision with the literature.

Guided by this information, researchers can better choose comparisons with the literature. In the manner of a more robust study.

Reviewer #3: 1. Why sample stop at 2021? Is there any influence in determining the PICC procedure by ultrasound in the era of the Covid pandemic?

2. Is there any data above 2022 that can be added for data updates?

3. Eldery criteria has not been listed

4. In discussion please elaborate more about infection related to PICC

5. What's the novelty about USG guided PICC

6. The results section does not yet discuss risk factors of PICC related infection

Reviewer #4: This manuscript presents a study of 161 patients undergoing ultrasound-guided PICC placement, with findings suggesting that accurate tip positioning is critical for reducing PICC-related infections. After revision, the manuscript shows notable improvements in grammar and background framing, which enhance both the readability and the credibility of the conclusions.

However, several minor but important issues still require attention. Addressing these points would further improve the rigor and overall quality of the manuscript.

1. Issue in the Abstract – Background Section

The background section of the abstract is currently inadequate. Instead of providing scientific context, it merely describes the study itself. A well-constructed background should briefly introduce the clinical significance of the topic and highlight the specific scientific problem the study aims to address. In this case, the authors should synthesize key points from the Introduction: namely, that PICC is a commonly used alternative to central venous catheterization, but its application carries risks—particularly catheter-related bloodstream infections—which warrant further investigation. This study seeks to explore how tip location, side of insertion, and catheter length may influence such complications. The abstract should be revised to reflect this rationale clearly and concisely.

2. Right-Side PICC Placement for Patients Without Prior PICC – Methodological Clarification

In the Methods section, the authors mention a preference for right-sided PICC placement in patients without prior PICC history. While this is a retrospective study and procedures have already been performed, it is important to clarify whether this decision was evidence-based. For example, literature supporting right-sided preference in internal jugular central venous catheterization—due to anatomical landmarks, vessel course, or operator handedness—may be relevant. If such methodological justification exists, it should be cited explicitly.

Furthermore, a potential bias may be introduced by choosing right-sided placement only for patients without prior PICC insertion, who may have had better baseline health conditions compared to those who required previous catheterization. The authors should discuss how such selection bias was addressed or mitigated in their analysis.

In conclusion, addressing these concerns would enhance the scientific integrity of the study and provide more robust evidence for optimizing ultrasound-guided PICC placement strategies to reduce complication rates related to suboptimal catheter positioning.

Reviewer #5: Thank you for the opportunity to review this manuscript. Unfortunately, I believe that the quality of this manuscript does not meet the standards of PLOS One, but this should not discourage the authors from continuing their valuable clinical research and thoroughly reviewing their manuscript before attempting submission elsewhere. Special attention should be given to better organisation of the structured abstract, and also to defining the research goals and hypothesis. It is unclear, but it looks like at least at one point the focus of this manuscript was infection control, in which case the developed protocol for infection control regarding placement, evaluation and maintenance of PICC should be discussed. Including an English language expert to proof-read the manuscript is advised.

**Do you want your identity to be public for this peer review?** For information about this choice, including consent withdrawal, please see our Privacy Policy

Reviewer #1: No

Reviewer #2: **Yes:** Felipe Martins Liporaci

Reviewer #3: No

Reviewer #4: No

Reviewer #5: No

---

## [Decision Letter · Decision Letter 2]

20 Oct 2025

Dear Dr. Lee,

**In addition to the reviewer comments below, please edit the Methods section of the Abstract to include more methodological items such as study dates, design, inclusion criteria, brief statistical analysis plans, etc. and the current information in the Methods of the Abstract can be condensed and perhaps moved to Results of the Abstract. As a minor point, please change "elderly" to "older adults."**

We look forward to receiving your revised manuscript.

Kind regards,

Kiyan Heybati

Academic Editor

PLOS ONE

Journal Requirements:

Reviewers' comments:

Reviewer's Responses to Questions

**Comments to the Author**

Reviewer #2: All comments have been addressed

Reviewer #6: All comments have been addressed

2. Is the manuscript technically sound, and do the data support the conclusions?

Reviewer #2: Yes

Reviewer #6: Partly

3. Has the statistical analysis been performed appropriately and rigorously?

Reviewer #2: I Don't Know

Reviewer #6: Yes

4. Have the authors made all data underlying the findings in their manuscript fully available?

Reviewer #2: Yes

Reviewer #6: Yes

5. Is the manuscript presented in an intelligible fashion and written in standard English?

Reviewer #2: Yes

Reviewer #6: No

Reviewer #2: An important topic that demands attention from everyone who performs invasive procedures. The authors reviewed and responded to all suggestions from the review team. Based on this, the work ready for publication.

Reviewer #6: Ultrasound is almost standard of care based on CDC and all other guidelines and should be used whenever available. The tip confirmation was still with CXR. So this is very standard operation.

https://www.cdc.gov/infection-control/hcp/intravascular-catheter-related-infections/summary-recommendations.html

Use ultrasound guidance to place central venous catheters (if this technology is available) to reduce the number of cannulation attempts and mechanical complications. Ultrasound guidance should only be used by those fully trained in its technique. IB

**Do you want your identity to be public for this peer review?** For information about this choice, including consent withdrawal, please see our Privacy Policy

Reviewer #2: **Yes:** Felipe Martins Liporaci

Reviewer #6: **Yes:** Siddharth Dugar

---

## [Editor Report · Decision Letter 3]

4 Nov 2025

Outcomes of a Bedside Ultrasound-guided Peripherally-inserted Central Catheter Placement across Critically-ill Older Patients

PONE-D-25-11583R3

Dear Dr. Lee,

We’re pleased to inform you that your manuscript has been judged scientifically suitable for publication and will be formally accepted for publication once it meets all outstanding technical requirements.

Kind regards,

Kiyan Heybati

Academic Editor

PLOS ONE
---

## [Editor Report · Acceptance letter]

PONE-D-25-11583R3

PLOS One

Dear Dr. Lee,

I'm pleased to inform you that your manuscript has been deemed suitable for publication in PLOS One. Congratulations! Your manuscript is now being handed over to our production team.

Kind regards,

on behalf of

Dr. Kiyan Heybati

Academic Editor

PLOS One